# Impact of Dietary Factors on Brugada Syndrome and Long QT Syndrome

**DOI:** 10.3390/nu13082482

**Published:** 2021-07-21

**Authors:** Sara D’Imperio, Michelle M. Monasky, Emanuele Micaglio, Gabriele Negro, Carlo Pappone

**Affiliations:** 1Department of Arrhythmology, IRCCS Policlinico San Donato, Piazza E. Malan 1, San Donato Milanese, 20097 Milan, Italy; Sara.DImperio@grupposandonato.it (S.D.); michelle.monasky@grupposandonato.it (M.M.M.); Emanuele.micaglio@grupposandonato.it (E.M.); gabriele.negro87@gmail.com (G.N.); 2Vita-Salute San Raffaele University, 20132 Milan, Italy

**Keywords:** Brugada syndrome, long QT syndrome, diet, ingredients, glucose, ketone bodies, ROS, sudden cardiac death

## Abstract

A healthy regime is fundamental for the prevention of cardiovascular diseases (CVD). In inherited channelopathies, such as Brugada syndrome (BrS) and Long QT syndrome (LQTS), unfortunately, sudden cardiac death could be the first sign for patients affected by these syndromes. Several known factors are used to stratify the risk of developing cardiac arrhythmias, although none are determinative. The risk factors can be affected by adjusting lifestyle habits, such as a particular diet, impacting the risk of arrhythmogenic events and mortality. To date, the importance of understanding the relationship between diet and inherited channelopathies has been underrated. Therefore, we describe herein the effects of dietary factors on the development of arrhythmia in patients affected by BrS and LQTS. Modifying the diet might not be enough to fully prevent arrhythmias, but it can help lower the risk.

## 1. Introduction

A healthy regime is fundamental for the prevention of cardiovascular diseases (CVD). CVD are widely known to be responsible for 33% of all deaths worldwide [1]. One of the multiple types of cardiac disorders is arrhythmia, which refers to a group of several conditions that interfere with heart rhythm. Among different types of cardiac arrhythmias, there are inherited channelopathies (IC), including long QT syndrome (LQTS), short QT syndrome (SQTS), Asian sudden unexplained nocturnal death syndrome (SUNDS), catecholaminergic polymorphic ventricular tachycardia (CPVT), and Brugada syndrome (BrS). Unfortunately, sudden cardiac death (SCD) could be the first symptom of patients affected by IC. Risk stratification and current diagnosis have been primarily focused on clinically detectable changes and abnormalities in the heart structure and function [2,3]. Several known markers are used to predict cardiac arrhythmias, although none are specific for patients affected by inherited channelopathies.

Several risks factors, such as obesity, diabetes, sleep apnea, anorexia nervosa, electrolyte imbalances, and unhealthy food consumption, are associated with these inherited channelopathies (Figure 1). Therefore, it is very important to address these risk factors in order to manage and prevent adverse outcomes. Most of the correlations are well established, and, indeed, these risks factors can be affected by adjusting lifestyle habits, such as a particular diet, impacting the risk of arrhythmogenic events and mortality. Thus, the aim of this review is to present objective insights into different daily diets. Specifically, here, we highlight the effects of various dietary factors and their suspected roles in the development of arrhythmias in BrS and LQTS, providing a summary of current literature and presenting questions that need to be the subject of future studies. This information is desperately needed to advise the patients diagnosed with these syndromes in the clinic. 

## 2. Brugada Syndrome and Long QT Syndrome

Brugada syndrome and long QT syndrome are among the most common inherited cardiac arrhythmias, with high risks of malignant arrhythmias, and they present with abnormalities in the 12-lead electrocardiogram (ECG): ST-segment elevation and prolonged QT interval, respectively. 

BrS is associated with right ventricular conduction abnormalities and coved-type ST-segment elevation in the right precordial leads of the ECG [4]. The syndrome is clinically characterized by syncope episodes and SCD due to ventricular fibrillation [5]. The majority of the patients are asymptomatic, and they are usually diagnosed by chance due to the dynamicity of the BrS ECG pattern, since it fluctuates throughout the day [6]. Moreover, these fluctuations can be induced by several factors, such as fever, hypercalcemia, diabetes, excessive consumption of food or alcohol, hyperkalemia, and certain drugs [7]. The coved-type ST-segment elevation is diagnostic when the “Type 1” Brugada pattern is seen, which can be found either on a spontaneous ECG or after a drug challenge with a class Ia anti-arrhythmic drug, such as ajmaline, which can provoke the Type 1 pattern so that the syndrome can be discovered [8]. The true incidence of BrS is currently unknown, due to the lack of availability of centers that can perform the drug challenge to unmask the syndrome. This is because these tests must be performed in specialized centers capable of specialized resuscitation due to the risks associated with the drug ajmaline [9], which can potentially induce life-threatening ventricular arrhythmias (VAs). For this reason, although used widely in Europe, the drug ajmaline is not used in the United States of America, due to risks associated with this drug. The BrS is believed to be genetic in nature, described as being oligogenic in nature, although it may occur in a minority of families as a Mendelian condition [10,11]. 

LQTS is defined by a variable degree of QT prolongation, absence of structural heart diseases, and it can manifest in three different subtypes (LQTS1, LQTS2, and LQTS3). The syndrome is clinically characterized by syncope episodes and SCD due to ventricular tachyarrhythmias [12]. LQTS can be congenital or acquired, therefore, a prolonged QT interval may result from genetic abnormalities, mineral imbalances, or certain medications [13]. It manifests as ECG abnormalities, including prolonged QT-interval, Torsade de Pointes (TdP), and ventricular fibrillation (VF) [14]. LQTS is usually diagnosed by measuring the QT interval on the ECG and it can include adrenalin and isoproterenol test. Moreover, epinephrine test is able to unmask the prolongation of the QT intervals in patients with ‘concealed LQTS’ [15]. The ECG may result normal at rest but based on the subtype, ECG abnormalities can be triggered by physical stress, emotional stress, dietary changes, certain drugs, or during sleep or at rest [16,17], and it has also been associated with metabolic syndromes and eating disorders [13]. LQTS primarily affects young people and is one of the main causes of SCD in this population [18]. 

## 3. BrS ECG Pattern Triggered by Food or Alcohol Intake

Several studies have repeatedly stressed that the BrS ECG pattern can appear after the consumption of alcohol [19,20] or caused by glucose-induced insulin secretion after the ingestion of a meal [21]. A 29-year-old BrS patient experienced several episodes of palpitation and syncope after alcohol consumption [22]. Another case report described a 25-year-old man who experienced three syncope events while consuming alcohol [23]. In yet another study, a 21-year-old man was diagnosed with BrS after he developed two episodes of syncope after the consumption of a high quantity of alcohol; specifically, his ECG, during the syncope, revealed ventricular fibrillation and the elevation of the ST-segment at high intercostal space [24]. In addition, a 56-year-old male experienced cardiac arrest while asleep after the consumption of one bottle of wine. The patient, five years previously, already had an abnormal ECG showing an elevation of the ST-segment, however, unfortunately, this abnormality was not considered important as a diagnostic criteria for BrS [25]. A nine-year-old boy, who was diagnosed with BrS by an ajmaline test, experienced syncope and cardiopulmonary arrest after the ingestion of a large hot dog [26]. Another case report of a 53-year-old man, who had been diagnosed with BrS after four weeks of Ramadan-fasting, had syncope events and several sudden cardiac arrests after the ingestion of a large meal [27]. 

Glucose and insulin intravenous infusion in patients affected by BrS results in a significant accentuation of the abnormal J-ST configuration [26]. In one study, 75% of BrS patients, compared to the controls, had a higher incidence of ECG fluctuation after an oral glucose tolerance test (OGTT) [28]. Although it is widely understood that large meals can trigger the BrS ECG pattern, the cellular mechanism responsible for the appearance of this ECG pattern by high levels of glucose and/or insulin concentration in BrS is still unclear.

In conclusion, the relationship between large meal intake, alcohol, and the development of the BrS ECG pattern is still unclear. However, it seems that the interplay between them is very relevant and should be investigated further.

## 4. Cortisol and Sudden Death

The majority of sudden cardiac deaths seem to occur from 4 am to 6 am, so in a period of nocturnal sleep in which cortisol concentration tends to be lower than in other periods of the day [29]. This observation suggests a role for abnormal sympathetic activity, which has been observed in a SUNDS cohort (sudden unexplained death during nocturnal sleep), and SUNDS is still considered to share common genetic causes with BrS [30,31].

Moreover, it is very well known that cortisol can affect the incidence and the clinical manifestations of sudden cardiac death. More recently, the role of the enzyme 11β-hydroxysteroid dehydrogenase (11β-HSD1) has been considered [32]. This enzyme is one of the most promising molecular targets to treat Type 2 diabetes mellitus and its complications [33]. Basically, 11β-HSD1 is able to catalyze the production of cortisol, and the levels of 11β-HSD1 correspond to the levels of cortisol concentration [29]. High concentrations of cortisol have already been demonstrated to be associated with both cardiac arrhythmias and diabetes mellitus [34]. The drug-mediated 11β-HSD1 inhibition alleviates most metabolic abnormalities associated with both diabetes and a cortisol concentration beyond normal levels [32]. The expression of 11β-HSD1 can be induced by high-fat diet in mouse models [35], suggesting the usefulness to assess the activity of 11β-HSD1 among human patients with BrS. Therefore, these mechanisms should be further explored to better understand why the majority of sudden deaths occur during this time period. 

## 5. The Mechanisms behind Food and Alcohol Intake as a Trigger for BrS ECG Pattern Manifestation

The mechanism behind the manifestation of ventricular arrhythmias after ingestion of alcohol or large amounts of food is still uncertain. The main product produced by the metabolism of fatty acids (FA), carbohydrates, ketones, and amino acids [36] is adenosine triphosphate (ATP), which is essential as energy source for cardiac work. The energy metabolic pathways include (1) glycolysis, where ATP is produced by glucose oxidation, (2) the citric acid cycle (Krebs cycle or TCA cycle), where guanosine-triphosphate (GTP), nicotinamide adenine dinucleotide (NADH) and flavin adenine dinucleotide (FADH_2_) are produced by acetyl-CoA oxidation, (3) the electron-transport chain (ETC), where most of the ATP is produced, and (4) fatty acid beta-oxidation, where FAs breakdown into acetyl-CoA and they are used by the TCA cycle. Acetyl-coenzyme A (CoA) is produced by either oxidative decarboxylation of pyruvate from glycolysis, beta-oxidation of long chain fatty acids, or oxidative degradation of some amino acids [37]. CoA is synthesized within the mitochondria, and it is the main substrate for the TCA cycle, which is a series of enzyme catalyzed reactions that generate energy sources via oxidative reactions. NADH and FADH_2_ are generated, then they enter into the ETC and, together with oxygen, they generate ATP through a redox reaction. The ATP level in the cells is maintained constant via two mechanisms: the production of ATP by oxidative phosphorylation and the hydrolysis of ATP [38]. In order to sustain normal cardiac activity in a healthy human heart, if there is an increase of fatty acid oxidation, there is also a decrease of pyruvate oxidation, and thus a glucose oxidation, and vice versa [39].

However, dysfunction of glucose metabolism, whether hypoglycemia or hyperglycemia, could have detrimental effects on cardiomyocytes. Several studies describe the deleterious effect of high glucose and alcohol on cardiomyocytes. In addition, the deprivation of glucose results in cardiac myocyte apoptosis [40,41,42]. A study conducted on rat cardiomyocytes demonstrated that high glucose-induced mitochondrial hyperpolarization increases cell injury [43]. Moreover, an in vitro induction of high glucose resulted in cardiomyocyte apoptosis [44]. Furthermore, several studies have demonstrated that ethanol depresses myocardial contractility both in humans and animals [45,46]. The first enzyme involved in alcohol metabolism is alcohol dehydrogenase (ADH), which catalyzes the conversion of ethanol into acetaldehyde, which is a reactive and toxic product that contributes to the formation of reactive oxygen species (ROS) and reduces the oxidation process in liver cells; the second enzyme involved is aldehyde dehydrogenase 2 (ALDH2), which converts acetaldehyde into acetate, which is then ready to be incorporated into Acetyl CoA and then entered into the TCA cycle. Indeed, a study conducted on 198 Japanese patients affected by BrS demonstrated that arrhythmic events caused by the consumption of alcohol were associated with the increased activity of the alcohol-metabolizing enzyme ADH1B in BrS patients [47]. 

BrS is a complex disease that has been described to have an oligogenic model of inheritance [10,48], and several possible genetic causes have been proposed, reviewed elsewhere [10,49]. It has also been hypothesized that, when people are overstressed by large meals and alcohol, especially during festivities, possible mutations in genes encoding for SULT1A enzymes result in the inability of those enzymes to deactivate catecholamines in the intestine, possibly inducing cardiac arrhythmia [50]. Finally, another hypothesis could be that glucose metabolism dysfunction can interfere with the homeostasis of ATP and ROS within the cardiomyocytes [51], possibly leading to arrhythmic events via mitochondria defects and impaired intracellular cation homeostasis. 

Adhering to the “Mediterranean diet” may reduce the risk of cardiovascular disease. However, unfortunately, this diet is not clearly defined. However, generally, it consists of eating smaller portions throughout the day, such as eating five smaller meals instead of three larger ones. The Mediterranean diet could allow for max one glass of red wine per day, but patients with cardiac arrhythmias are probably best to abstain completely from alcohol, especially if even low amounts of alcohol make them feel unwell. In our experience, even one glass of alcohol can make some patients with BrS fell unwell. In these cases, it is best to avoid alcohol completely. The Mediterranean diet includes a lot of plant-based foods and olive oil, but tends to be low in saturated fat, meat, and dairy products. Red meat could be eaten once per week. Fish are generally included minimum twice per week, but oily fish, such as salmon, which are larger fish and could contain a higher amount of mercury, should be limited to only once per week, although it is still good because it contains a high amount of omega-3 fatty acids. Nuts are also good for omega fatty acids and legumes are good for proteins. 

## 6. Sudden Cardiac Death and QT Prolongation Triggered by Ketogenic Diet

While carbohydrates provide a readily available fuel for our body, fats and oils (lipids) are considered our primarily source of stored energy. Fat enters into the body through food and breaks down into triglycerides, and then into fatty acids and glycerol. Mitochondria provide the main source of energy, and, therefore, the dysfunction of the two metabolic pathways of β-oxidation of fatty acids (FAO) and oxidative phosphorylation (OXPHOS) can lead to mitochondrial remodeling and the manifestation of heart failure, arrhythmias, and ventricular hypertrophy [52,53].

Acute arrhythmic events associated with a ketogenic diet and calorie-restricted diets can include QT prolongation, leading to sudden cardiac death, and these phenomena have been described in children, adolescents, and adults [54,55,56]. A ketogenic diet is characterized by very low carbohydrate intake, with 75% of calories derived from fat. This carbohydrate (CHO) restriction helps to reduce blood glucose and insulin [57]. However, it is known that, among several complications related to a ketogenic diet, there is selenium deficiency, QT prolongation, and SCD [58,59,60]. A case report described a Torsades de Point (TdP) event in a patient with a dual-chamber implantable cardioverter-defibrillator (ICD) affected by LQT2 triggered by an uncommon factor: the ketogenic diet. Specifically, while the patient was following a ketogenic diet, she experienced four episodes of ventricular fibrillation, due to TdP, over the course of only three weeks. Selenium, ketone bodies, and alcohol levels were all within normal limits. One month later, she stopped the ketogenic diet, and the ICD showed no further arrhythmic episodes [17]. 

The majority of SCD cases of pediatrics patients are associated with QT interval prolongation and ketosis. In a report about two cases of death in two children on a ketogenic diet for seizure control, both patients experienced QT prolongation and suffered from selenium deficiency [58]. However, another case described a correlation between a QT interval prolongation and ketogenic diets in the absence of electrolyte imbalance in children [60]. Specifically, a direct correlation was observed between QT interval prolongation and β-hydroxybutyrate concentrations, and between QT interval prolongation and systemic acidosis [60]. Interestingly, in a study of 70 children with drug-resistance epilepsy, receiving a ketogenic diet for a 12-month period, no deleterious effects on corrected QT interval, QT dispersion, and Tp-e interval were reported [61]. Finally, another case described a five-year-old boy who developed selenium deficiency, acute reversible cardiomyopathy and ventricular tachycardia with prolonged QT interval after following a ketogenic diet to treat refractory epilepsy. Then, his clinical status improved and got back to normal after selenium supplementation [62]. 

In conclusion, the extended association between QT interval prolongation and/or SCD and ketosis conditions/ketogenic diet has been described, and it raises the question of whether ketosis may directly affect cardiac repolarization. 

## 7. The Mechanisms behind Ketosis as a Trigger for the Manifestation of QT Prolongation 

The relationship behind the manifestation of ventricular arrhythmias and high-fat and low-carbohydrate diet is still unclear. However, it is known that the heart uses fatty acids as a main substrate for source energy, but it can also use ketone bodies. Ketone bodies are produced by breaking down fatty acids and ketogenic amino acids in a process called ketogenesis, previously reviewed elsewhere [63]. Briefly, ketogenesis involves the anabolic hormone insulin, and the catabolic hormones glucagon, cortisol, catecholamines, and growth hormone, of which insulin and glucagon are considered the most crucial for this pathway [64,65] (Figure 2). Ketone body concentration is lowered by insulin, which promotes glucose uptake and oxidation. The reduction of circulating insulin levels is the principal triggering event for accelerating ketogenesis [63]. Insulin acts on the adipose tissue, liver, and the periphery; specifically, the low amount of insulin and a high amount of glucagon in our blood stream trigger the augmentation of free fatty acids (FFAs), increase uptake of FFAs into the mitochondria, and increase production of ketones in the liver, by activating the acyltransferase system through the inhibition on malonyl-CoA synthesis. Indeed, the ketogenesis pathway occurs primarily in the mitochondria of hepatocytes. FFAs are converted into fatty acyl CoA (Acyl CoA), which enters into the hepatic mitochondria through CPT1-mediated transport. Then Acyl CoA undergoes β-oxidation to produce Acetyl CoA, which is only employed to generate ATP if there is enough oxaloacetate. When carbohydrate intake is limited, such as in the ketogenic diet, the liver uses the majority of oxaloacetate to produce glucose through gluconeogenesis; therefore, the liver diverts the acetyl CoA to form ketone bodies. Thiolase enzyme (acetyl coenzyme A acetyltransferase (ACAT)) catalyzes the reversible reaction where two molecules of acetyl CoA are combined to generate acetoacetyl CoA. At this point, mitochondrial β-Hydroxy β-methylglutaryl-CoA (*HMG*-CoA) synthase catalyzes a condensation reaction by adding an extra acetyl CoA molecule onto the acetoacetyl CoA. Then, the enzyme *HMG*-CoA lyase cleaves the *HMG*-CoA, which releases CoA and forms acetoacetate, the metabolized ketone body. Within the mitochondrial matrix, 3-hydroxybutyrate dehydrogenase can reduce the acetoacetate into two other ketone bodies: acetone and β-hydroxybutyrate (β-HB), through non-enzymatic decarboxylation or by beta-hydroxybutyrate dehydrogenase, respectively. Moreover, the ratio of NADH/NAD^+^ helps to maintain an equilibrium between acetoacetate and β-HB within the mitochondria matrix. Both acetoacetate and β-HB are considered fuel molecules normally found in the heart and renal cortex. Acetone cannot be metabolized. At this point, due to the fact that the liver does not have enzyme beta ketoacyl-CoA transferase, and it cannot utilize ketone bodies [66], acetoacetate and β-HB reach the extrahepatic tissues. β-HB is converted into acetoacetate, which is then converted back to acetyl-CoA. Acetyl-CoA goes through the TCA and produces 22 ATP by oxidative phosphorylation. Therefore, due to the acid nature of ketone bodies, this causes an anion gap metabolic acidosis. This condition usually results in electrolyte imbalances, especially a reduction in K^+^, Mg^2+^, and P. 

A metabolomic study on individuals with arrhythmogenic cardiomyopathy (AC) identified as a possible biomarker β-HB, due to its elevated amount in the plasma and hearts [67]. Specifically, the β-HB produced by cardiomyocytes of AC patients is released into the blood, and its levels are significantly higher compared to controls [67]. Therefore, it was demonstrated that cardiac ketogenesis occurs in CA, and β-HB may be used as a potential metabolic marker to predict CA.

## 8. Oxidative Stress

Oxidative stress is usually considered a state in which the production of ROS and antioxidant defenses are not balanced [68]. ROS are derivatives of molecular oxygen, such as superoxide (O_2_^−^), hydrogen peroxide (H_2_O_2_), peroxynitrite (ONOO^–^), and hydroxyl radicals (OH) [69]. ROS are mainly produced by mitochondria, and their homeostasis is maintained by the enzyme glutathione peroxidase (GSH-Px). In damaged mitochondria, the calcium ions are overloaded and drive the augmentation of ROS concentration, which leads to excitotoxicity damage [70]. Indeed, excessive generation of ROS, impaired calcium homeostasis, and diminished ATP production directly impact mitochondrial function [71].

Repetitive or prolonged oxidative stress can damage proteins and lipids within the cell, and it might result in a contractile dysfunction, a downregulation of gene expression, and also a disrupted energy transfer, which could induce cardiomyocyte apoptosis, followed by heart failure [72]. There have been limited studies into the mechanisms linking oxidative stress and arrhythmias, but it has been shown that cardiac conditions with increased arrhythmic risks are associated with an unbalanced production of ROS [73]. ROS, in addition to their role as a messenger in cell signal transduction and the cell cycle, regulate both cellular metabolism and ion homeostasis in excitable cells. An elevated presence of ROS within the cells can be highly toxic, and they can lead to arrhythmogenic triggers, such as alterations of ion channels (Na, Ca^2+^, and K^+^), dysfunction of the mitochondria, and gap junction remodeling [74]. Therefore, an excessive production of ROS and ineffective ROS scavenging can culminate in cell death [52].

It is well known that ketone bodies are always present in the blood, and their concentration increases during prolonged exercise and fasting. In vitro studies demonstrated that KBs stimulate insulin release [75,76], cause lipid peroxidation, and generate oxygen radicals [77]. It is also known that KBs are able to reduce oxidative stress, due to the activation of multiple protective antioxidant pathways. However, we hypothesize that patients affected by inherited channelopathies could have a potential dysfunction of these pathways, and they are not able to reduce the levels of ROS. To sustain our hypothesis, there are some studies suggesting that exposure to high concentrations of KBs could provoke oxidative stress. A study conducted on calf hepatocytes suggested that both β-HB and acetoacetic acid decrease the activity of antioxidant enzymes superoxide dismutase (SOD), catalase (CAT), and glutathione peroxidase (GSH-Px), and increase malondialdehyde (MDA) and nitric oxide (NO), which are markers of oxidative stress [78,79]. Another study attributed acetoacetate treatment to the activation of mitogen-activated protein kinase (MAPK) pathway, which is known to be activated by oxidative stress in rat hepatocyte cells [80]. Therefore, even if there are several studies associating KBs with the inhibition of oxidative stress and ROS production, there are still some studies showing a correlation between KBs and the induction of oxidative stress. 

It is known that a nutrient overload helps to release free fatty acids and can also induce damages to the mitochondria [81]. Therefore, free fatty acids might be related to the excess production of oxidative stress. Indeed, there is a lot of evidence suggesting that conditions of high levels of glucose, lipids, or their combination can interfere with mitochondria metabolism, and they may modulate mitochondrial ATP synthesis capacity and increase ROS production [71]. Moreover, in patients with inherited channelopathies, malignant ventricular arrhythmias and SCD occur before overt structural changes of the heart. To prevent the arrhythmogenic substrate progression, more studies to understand the electrical instability of cardiomyocytes are needed. At present, a lot of studies are focused on the connection between metabolic disease and the manifestation of arrhythmic events. Specifically, the attention is focused on the mitochondrial dysfunction, which can drive the manifestation of arrhythmic events by interfering with the electrical activity of the cardiomyocytes. The role of mitochondrial dysfunction in inherited channelopathies is still unclear and will be the subject of future studies.

## 9. Vitamin D

Vitamin D, also known as calciferol or hydroxyvitamin D (25-OHD), is a fat-soluble vitamin that is naturally found in a few foods and is also produced endogenously by the exposure of the skin to sunlight ultraviolet rays [82]. Vitamin D is responsible to increase the intestinal absorption of calcium and to maintain the serum calcium and phosphate concentration [82]. Vitamin D is also involved in the reduction of inflammation, cell growth, neuromuscular, immune function, and glucose metabolism [83,84,85]. 

Vitamin D deficiency has been related to different cardiovascular disorders, including SCD [86]. Moreover, a decreased level of 25-OHD has been linked to structural and ionic channel remodeling, which may increase arrhythmic events [87]. Indeed, prolonged QTc is commonly induced by hypocalcemia, which can be caused by vitamin D inadequacy or resistance. 

In a case of a patient with severe vitamin D deficiency, hypocalcemia and prolonged QTc resulted in TdP and cardiac arrest. After the administration of vitamin D and calcium supplements, the QTc interval became normal, and the patient did not experience additional arrhythmic events [88]. A hypocalcemic teenage girl affected by hypoparathyroidism experienced a few episodes of syncope during exercise, and the ECG on admission showed prolonged QTc. After the treatment with alphacalcidol, which is an analog of vitamin D, and calcium supplements, the QTs of the patient became normal [89]. Moreover, in a case of a 40-year-old woman, who followed a vegan diet, she was affected by hypocalcemia, due to severe deficiency of vitamin D. The patient manifested symptoms of palpitations, presyncope, and a long QT (556 ms). Therefore, she was first treated with calcium gluconate, then by vitamin D and calcium oral supplementation. After the treatment, the QTc normalized, and the symptoms disappeared [90].

Vitamin D deficiency can not only induce the prolongation of the QTc, but it can also promote inflammatory reactions. It is known that cardiac contraction is affected by an overload of Ca^2+^ ions in myocardial cells. Studies have hypothesized that lack of vitamin D could interfere with the function of Ca^2+^ in myocardial cells; specifically, it can induce hypertrophy [91], increase anti-inflammatory cytokines [92], increase fibrosis [93], and impact the production of ROS in the atria [94]. 

The correlation between atrial fibrillation and vitamin D is poorly understood. Some studies suggest a direct correlation between atrial fibrillation and vitamin D deficiency [95,96,97]. However, other studies did not find a connection between vitamin D levels and the incidence of atrial fibrillation [98,99,100]. It would be important to better understand the link between vitamin D and calcium, because it is known that calcium overload causes myocytes apoptosis and cardiac failure [101,102]. Oxidative stress associated with cardiovascular risks factors is real, and, therefore, further studies are needed to investigate this aspect. 

In conclusion, vitamin D deficiency appears to play an important role in CVD, and it is important to investigate the possible role of vitamin D deficiency in the development of arrhythmic events.

## 10. Omega-3 Fatty Acids

Omega-3 fatty acids, also called ω-3 (n-3) fatty acids, are polyunsaturated fatty acids (PUFAs), and they are essential nutrients involved in lipid metabolism. It has been demonstrated that n-3 PUFAs, including eicosapentaenoic acid (EPA), docosahexaenoic acid (DHA), and α-linolenic acid (ALA), play an important role in the human diet and cellular physiology, and they may have beneficial effects against CVD and risks factors, including arrhythmias, probably by the modulation of cardiac ion channels [103]. DHA and EPA fatty acids are mostly found in seafood, seaweed, and algae, while ALA is in nuts and seeds. ALA might improve cardiac function by inhibiting apoptosis through anti-inflammatory and anti-oxidative stress effects in diabetic, but not normal, rats [104]. 

An investigation into whether a diet enriched with fish and PUFAs could be associated with changes in QT duration on a resting ECG showed that long-term consumption of fish could positively influence the duration of QTc by lowering it; therefore, fish intake may be considered an antiarrhythmic protection [105]. A retrospective study of men affected by BrS evaluated the correlation between the serum levels of EPA and DHA and the risks factors for SCD. A multivariate logistic regression analysis showed that low levels of EPA and DHA were linked to the incidence of syncope in patients affected by BrS. This same study suggested that all levels of omega-3 PUFAs may play an important role in preventing ventricular fibrillation in BrS [106]. Furthermore, a study on 123 Langendorff-perfused rabbit hearts, used to mimic LQT2 and LQT3 syndrome, showed that PUFAs were able to prevent TdP by reverting the AP prolongation. This effect was stronger in LQT2 than in LQT3 syndrome, and the antitorsadogenic effect was more distinctive with DHA and EPA compared with ALA [107]. 

A recent review demonstrated that n-3 PUFAs have no significant effect on mortality or cardiovascular health [108]. For example, in a study investigating the correlation between n-3 PUFAs from fish and risks of CVD, including SCD, PUFAs were inversely related to QTc and JTc intervals. However, QTc and JTc did not reduced the inverse relationship between n-3 PUFAs and SCD risks, suggesting that this association cannot explain the prevention of prolonged ventricular repolarization [109]. Moreover, in a study investigating the association between mercury (Hg), EPA, and DHA, and large seafood consumption and heart rate variability (HRV) and QT interval duration, the authors found a possible association between specific seafood types and arrhythmias, such as tuna steak with QTc and anchovies with HRV [110]. Finally, data from four trials suggested that a high dose (4.0 g/d) of n-3 PUFAs could increase the risk of atrial fibrillation development [111]. 

In conclusion, data on the effects of omega-3 are still inconsistent, requiring further studies to assess their beneficial effects on preventing arrhythmic events. 

## 11. Arrhythmogenic Ingredients

Several factors can increase the risk of the development of arrhythmias in BrS and LQTS. Although not well studied in these two specific syndromes, one of the factors that could result in arrhythmias is the consumption of specific foods, which are considered to be arrhythmogenic [112,113], as described in Table 1, which lists both arrhythmogenic and anti-arrhythmogenic dietary factors.

Several case reports have described patients experiencing arrhythmic events and cardiac arrests after the consumption of energy drinks [114,115,116,117]. Taurine has been shown to modulate ion channel activity by suppressing the activity of sodium, calcium, and potassium channels [118]. Moreover, it shortens the action potential duration and decelerates the rate of terminal repolarization of the cardiac action potential, inducing atrial and ventricular arrhythmias or cardiac arrest [116]. Furthermore, atrial and ventricular arrhythmias can also be induced by caffeine, which is able to interfere with calcium homeostasis [115,119] by increasing intracellular calcium concentration [115]. Indeed, it was demonstrated that caffeine is able to stimulate calcium release from the sarcoplasmic reticulum [120], and calcium imbalances, particularly sarcoplasmic reticulum calcium stores, may be altered in BrS [121,122,123]. A 24-year-old male developed arrhythmias and collapsed after the ingestion, for the first time, of a small quantity of an energy drink combined with alcohol; specifically, the drink contained 80 mg of caffeine and 1000 mg of taurine, plus Vodka. He was then diagnosed with BrS [116]. Another case report showed that energy drinks could have triggered an abnormal QT response in a 13-year-old female affected by LQTS1. Moreover, a 22-year-old female affected by LQTS1 experienced cardiac arrest after the consumption of an elevated quantity of energy drink [124]. 

Grapefruit has been identified as an inhibitor of drug-induced TdP and QTc prolongation, thus enhancing their pharmacokinetics [113]. Specifically, naringenin, which is a flavonoid found in grapefruit, is able to block the hERG channel and induce TdP and/or QTc prolongation [113,125]. These hERG channels are known receptors of the drug ajmaline, which is used to provoke the diagnostic type-1 BrS ECG pattern [126]. Therefore, other ingredients could be suspected to be arrhythmogenic, because they contain flavonoids: citrus fruit, parsley, onions, berries, bananas, red wine, chocolate, grains, nuts, tea, coffee, and various other fruits and vegetables [127,128,129], including spinach, cauliflower, broccoli, black beans, and chickpeas [130,131,132,133,134]. Moreover, additional ingredients like lemon, lime, clementine, oranges, and bergamot oil could be considered a risk, due to the same organic compounds of grapefruit: furanocoumarins [129]. However, other studies have suggested that flavonoids could actually be beneficial for cardiovascular health by reducing inflammation. Flavonoids, polyphenolic compound derivatives, reduce inflammation and risk of cardiovascular disease by reducing NFκB and its resulting transcription factors involved in the inflammatory pathway [128]. Many other natural products, which have polyphenols as their major compound, have been shown to have anti-inflammatory effects, such as mushrooms, honey, plant extracts, plant juices, plant powders, and essential oils [128,135,136]. Thus, further studies are needed to understand these foods and the resulting effects, in order to better advise the patients about dietary supplements or restraints. 

## 12. Ingredients That Suppress Cardiac Arrhythmogenesis

An interesting review showed that 18 active ingredients such as alkaloids, flavonoids, saponins, quinones, and terpenes, Wenxin-Keli, and Shensongyangxin could have antiarrhythmic effects [143]. In particular, the Chinese herb extract, Wenxin-Keli (WK), has been reported as an effective treatment of atrial and ventricular fibrillation by inhibiting the transient K^+^ outward current (I_to_) [144]. The experiments were conducted in a canine experimental model of BrS, and the authors observed an inhibition of I_to_ and indirect adrenergic sympathomimetic effects using WK in combination with quinidine [144]. However, the 18 ingredients described in the review, with the exception of omega 3, have been tested in vitro or in animal models as natural drugs or in combination with other antiarrhythmic treatments, and not as ingredients for regular meals. Therefore, further studies are needed. 

Another ingredient with the potential to prevent antiarrhythmic events, in moderate doses, is resveratrol, a stilbenoid polyphenol found in grapes. It has been described as a potential inhibitor of intracellular calcium release able to eliminate calcium overload in AF, and, therefore, able to preserve the cardiomyocyte contractile function [140]. Finally, based on an interesting review on antioxidant therapies for atrial fibrillation [138], it would be interesting to further study the role of vitamin C, vitamin E, and carotenoids, found in fruits and vegetables, and their role as suppressors of ROS, and, therefore, their role as anti-arrhythmic nutrients. 

## 13. Electrolytes

Food is the main source of electrolytes, essential minerals for life. Electrolytes are required for maintaining osmotic pressure in cells and generating action potentials in nerves and muscles. In particular, sodium, calcium, potassium, and magnesium play an important role in the heart. Indeed, it has been demonstrated that electrolyte imbalance can have detrimental effects on the heart, such as triggering cardiac arrhythmias or cardiac arrest [141], including what has been coined “BrS phenocopies” [145], which may provide important insights into the mechanisms involved in BrS [11]. It is important to understand better the connection between food intake, electrolyte imbalance, and BrS or LQTS. 

## 14. Vagal Tone Activity and Arrhythmic Events 

The heart rhythm is regulated by cardiac parasympathetic (vagal) nerves, the sympathetic nerves, and the pacemaker cells [121]. Autonomic activity can influence the elevation of the ST-segment [146,147,148]. Indeed, late in the 1990s, it was shown that the nocturnal vagal activity may be involved in the cardiac arrhythmic events of BrS [149]. Moreover, the high vagal activity could lead to the manifestation of ventricular tachyarrhythmias in patients with BrS or LQTS [150,151].

The relationship between the autonomic modulation and arrhythmic events is very complex and still unclear. However, based on the studies presented in this review, the association between the vagal activity and the consumption of large meals may contribute to the manifestation of arrhythmic events. 

## 15. Conclusions

We describe herein the effects of dietary factors in patients affected by cardiac arrhythmias, specifically BrS and LQTS. To date, the importance of understanding the relationship between diet and inherited channelopathies has been underrated. It is evident that dietary factors can influence the risk of the development of arrhythmic events. Therefore, we recommend eating and drinking small portions throughout the day and to try to limit certain types of ingredients in order to prevent arrhythmic events. Modifying the diet might not be enough to fully prevent arrhythmias, but it can help lower the risk.

## Figures and Tables

**Figure 1 nutrients-13-02482-f001:**
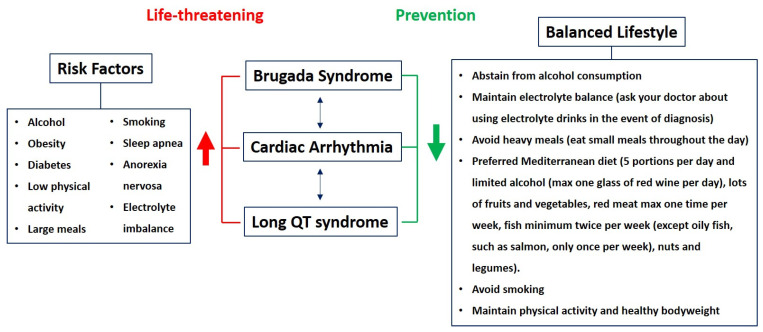
Cardiac arrhythmias, such as Brugada syndrome and long QT syndrome, can be triggered by a variety of factors, but maintaining a certain lifestyle may reduce these risks.

**Figure 2 nutrients-13-02482-f002:**
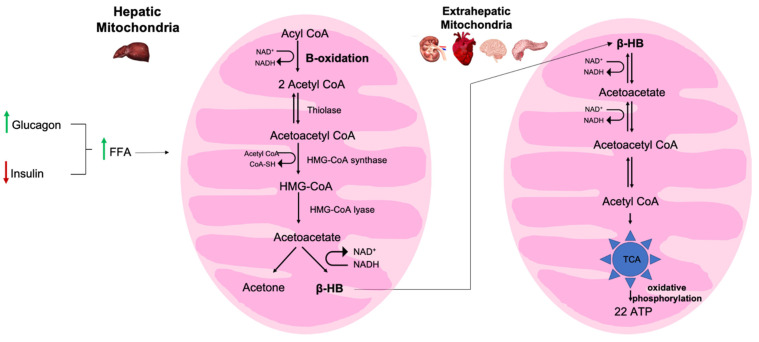
Ketogenesis pathway in hepatic mitochondria. Low amounts of insulin and high amounts of glucagon in our blood stream trigger the augmentation of free fatty acids (FFAs), and increase uptake of FFAs into the mitochondria. Acetyl CoA is generated from beta-oxidation of Acyl CoA, which derives from FFAs. The production of ketones (acetoacetate, acetone, β-HB) in the liver is then increased.

**Table 1 nutrients-13-02482-t001:** Arrhythmogenic and anti-arrhythmogenic dietary factors.

Possible Dietary Triggers Involved in Cardiac Arrhythmia	
Pro-Arrhythmic	Anti-Arrhythmic
Alcohol [22,23,24,137]	Vitamin D, E, C [138]
Heavy Meals [21,27,139]	Carotenoids [138]
Energy Drinks [114,115,116,117]	Resveratrol [140]
Flavonoids (found in grapefruit [125], citrus fruit [113,125,129], parsley [129], onions, berries [129], bananas, red wine [129], chocolate [129], grains, nuts, tea, coffee, spinach, cauliflower, and broccoli)	Electrolytes [141]
Licorice [142]	Omega–3 [103,105,106]
Honey [113,135] ^§^	Honey [127,128,135,136] ^§^
Mushrooms [113] ^§^	Mushrooms [136] ^§^

^§^ Honey and mushrooms have been described as both arrhythmogenic and anti-arrhythmogenic, depending on the study.

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
