# Peer review of "Impact of Dietary Factors on Brugada Syndrome and Long QT Syndrome"

_nutrients, 2021, doi:10.3390/nu13082482_

Round 1
Reviewer 1 Report
Pag. 2: BrS is associated with right ventricular conduction abnormalities and coved-type ST- segment elevation in the right precordial leads (Monasky et al., 2020).
Comment: This reference enhance the importance of a junior doctor but does not reflect the true history of the syndrome
Pag. 2: The coved-type ST-segment elevation is diagnostic when the “type 1” Brugada pattern is seen, which can be found either on a spontaneous ECG or after a drug challenge with a class Ia anti-arrhythmic drug, such as ajmaline.
Comment: This sentence, despite somewhere accepted also from some consensus, has not enough strength to be considered an evidence based approach to the syndrome. An ECG is not a syndrome, especially if it is drug induced. A routinary drug challenge in healthy subjects with a nearly normal ECG is not an ethical practice.
Pag. 3: plasma glucose and insulin concentration increase after the ingestion of a meal or alcohol and may interfere with ST-segment elevation (Nogami et al., 2003
Comment: this is mainly an interesting experimental observation. Glucose and insuline can modify ST segment elevation indeed but do not induce other ECG patterns, namely J point modification or late qrs fractioning
Pag 3: Moreover, it is also very important to study the mechanisms behind ethanol metabolism and cardiac outcomes
Comment: the association between alcohol intake and the syndrome is and limited to rare case reports that cannot evidence a strict correlation between the two conditions. “The holiday heart syndrome”, that causes cardiac arrhythmias after the week end is very well known cause of different arrhythmia, not only those linked to a rare syndrome. The recent paper by Wu (2019) discuss that experimentally the arrhythmic events after alcohol drinking was associated with enhanced activity of alcohol‐metabolizing enzyme ADH1B, but of course it is so difficult to state that patients with the syndrome have also this enzyme abnormality.
Thus, both excessive glucose and alcohol intake are not an healthy food for any human, and surely those with a cardiac disease are prone to adverse effect. This is a general adverse effect, and not an evidence based specific sign for some distinct syndrome. Gastric discomfort may also induce an abnormal vagal drive that can be the true responsible both for the st elevation and the arrhythmias
Page 5: Literature suggests that the majority of SCD cases of pediatrics patients are associated with QT interval prolongation and ketosis
Comment: this sentence need evidence based data.
Page 9: In conclusion, vitamin D deficiency appears to play an important role in CVD, and it is important to investigate the possible role of vitamin D deficiency in the development of arrhythmic events.
Comment: this an author’s opinion, reflecting inconclusive data
Page 10: ]. Specifically, naringenin, which is a flavonoid found in grapefruit, is able to block the hERG channel and induce TdP and/or QTc prolongation (RaymondL.Woosley, 2020; Scholz et al., 2005). These hERG channels are known receptors of the drug ajmaline, which is used to provoke the diagnostic type-1 BrS ECG pattern [133]. Therefore, other ingredients could be suspected to be arrhythmogenic, because they contain flavonoids: citrus fruit, parsley, onions, berries, bananas, red wine, chocolate, grains, nuts, tea, coffee, and various other fruits and vegetables
Comment: so many dangerous food. What can we eat? Possibly flavonoids are of some danger for a so limited number of people with enzymatic abnormalities. Only in the far future it will be possible to clarify these assumptions
Reviewer 2 Report
The manuscript entitled “Impact of Dietary Factors on Brugada Syndrome and Long QT Syndrome”, by D’Imperio and colleagues, reviews the effects of dietary factors on the development of arrhythmias. The manuscript provides an overview of the evidences that support the link between dietary factors and risk of arrhythmias, focusing on patients diagnosed on BrS and LQTS, and suggests that modifications in the diet might be crucial to prevent cardiac arrhythmias. This is an interesting and relevant topic, although some aspects of the review should be further improved.
Major criticisms:
1- To explain how dietary factors could protect or trigger arrhythmias, the authors start every section with an introduction about metabolism that often is too basic and long (for example, the introduction on glucose metabolism in pages 2-3). Shorter and clearer overviews and a focus on the the links with arrhythmias would facilitate the reading of the review.
2- The article misses some criticism about the articles mentioned. For some evidences, there is only “association” or “correlation”, which is not evidence of cause. The authors should differentiate those studies that are solid and well-designed from those studies that show weak evidences and focus on solid evidences that are strong and with mechanistic insights.
3- The two figures of the review are basic (especially Figure 1) and focused on metabolism. I suggest to make more comprehensive figures and include the potential link to the effects on arrhythmias.
4- A general comment is that the authors, in some sections, focus on the effects of diet on BrS and LQTS described, and in other sections just describe the role of dietary factors on CVD, which is too broad. Since the emphasis of this review is that nutrients and diet can affect arrhythmia events in BrS and LQTS patients, they should concentrate on the evidences linking to these inherited arrhythmias.
5- In page 4, the paragraph describing the TCA cycle and energy production is not clear. Of note, either in the text or the figure, it should be taken into consideration that some amino acids can also enter the TCA cycle through different intermediates, not only through Acetyl-CoA.
6- Ketogenesis and its hormonal regulation (pages 5-6) should be better described in the text. Not clear also the role of ketogenesis for the brain and other organs during fasting. Of note, the sentence “The ketogenesis involves the anabolic hormone insulin, and the catabolic hormones glucagon, cortisol, catecholamines, and growth hormone, of which insulin and glucagon are considered the most crucial for this pathway” is not very specific.
7- Table 1 is interesting but misleading, since some nutrients that are in both columns are also in different lanes. I suggest the authors to make a clearer table for the reader.
Minor criticisms:
1- The authors should revise the format of the references throughout the text.
2- The authors should also revise abbreviations, which sometimes are not defined the first time they appear.
